# The roles of MRI-based prostate volume and associated zone-adjusted prostate-specific antigen concentrations in predicting prostate cancer and high-risk prostate cancer

Song Zheng[1☯], Shaoqin Jiang[1☯], Zhenlin Chen[1☯], Zhangcheng Huang[1☯], Wenzhen Shi[1], Bingqiao Liu[1], Yue Xu[1], Yinan Guo[2], Huijie Yang[1], Mengqiang Li[1]*

1 Laboratory of Urology, Department of Urology, Fujian Union Hospital, Fujian Medical University, Fuzhou, Fujian, China, 2 Department of Nursing, Laboratory of Urology, Department of Urology, Fujian Union Hospital, Fujian Medical University, Fuzhou, Fujian, China

☯ These authors contributed equally to this work.
* limengqiang1976@163.com

## Abstract

Prostate biopsies are frequently performed to screen for prostate cancer (PCa) with complications such as infections and bleeding. To reduce unnecessary biopsies, here we designed an improved predictive model of MRI-based prostate volume and associated zone-adjusted prostate-specific antigen (PSA) concentrations for diagnosing PCa and risk stratification. Multiparametric MRI administered to 422 consecutive patients before initial transrectal ultrasonography-guided 13-core prostate biopsies from January 2012 to March 2018 at Fujian Medical University Union Hospital. Univariate and multivariate logistic regression analyses and determination of the area under the curve (AUC) of the receiver operating characteristic (ROC) curve was performed to evaluate and integrate the predictors of PCa and high-risk prostate cancer (HR-PCa). The detection rates of PCa was 43.84% (185/422). And the detection rates of HR-PCa was 71.35% (132/185) in PCa patients. Multivariate analysis revealed that prostate volume(PV), PSA density(PSAD), transitional zone volume(TZV), PSA density of the transitional zone(PSADTZ), and MR were independent predictors of PCa and HR-PCa. PSA, peripheral zone volume(PZV) and PSA density of the peripheral zone(PSADPZ) were independent predictors of PCa but not HR-PCa. The AUC of our best predictive model including PSA + PV + PSAD + MR + TZV or PSA + PV + PSAD + MR + PZV was 0.906 for PCa. The AUC of the best predictive model of PV + PSAD + MR + TZV was 0.893 for HR-PCa. In conclusion, our results will likely improve the detection rate of prostate cancer, avoiding unnecessary prostate biopsies, and for evaluating risk stratification.

**Data Availability Statement:** All relevant data are within the manuscript and its Supporting Information files.

**Funding:** This work was supported by Startup Fund for scientific research, Fujian Medical University (Grant number: 2016QH032), Fujian Natural Sciences Foundation (Grant number: 2017J01203) and Joint Funds for the innovation of science and Technology, Fujian province (Grant number: 2017Y9023). The funders had no role in study design, data collection and analysis, decision to publish, or preparation of the manuscript.

**Competing interests:** The authors have declared that no competing interests exist.

## Introduction

PCa is the most frequent cancer in men[1], with increasing prevalence.[2] Screening of PSA can detect PCa at an earlier stage. However, elevation of PSA levels in serum requires prostate biopsy to confirm if it is caused by PCa. Unfortunately, a biopsy can be painful and may cause complications such as infection and bleeding.[3] Fewer than 50% of patients with elevated PSA levels have positive biopsies (41.49%[4] and 30.7%[5]). The low detection rate is partly explained by the blind approach of transrectal ultrasound scan (TRUS)-prostate biopsy,[6] which leads to a high rate of unnecessary biopsies.

PSA is secreted by normal and malignant prostate tissues. It follows therefore that PSA is an organ-specific rather than a cancer-specific serum marker, which means that the elevation of PSA levels in patients with negative biopsies can be caused by benign prostatic hyperplasia (BPH) and prostatitis.[2] Compensating for the limitations of PSA tests is achieved by adjusting PSA levels according to prostate volume (PV), known as PSA density (PSAD).[7, 8]

According to MRI imaging of prostate zonal anatomy, the prostate comprises a peripheral zone (PZ), a transition zone (TZ), a central zone, and an anterior fibromuscular stroma.[9] The PZ is the source of 75% to 85% of PCa.[10] Compared with PSA levels alone, the accuracy of diagnosing PCa will be improved using PZ- adjusted PSA levels (PSA density of peripheral zone [PSADPZ]), derived from the ratio of PSA and peripheral zone volume, or TZ-adjusted PSA levels (PSA density of transition zone [PSADTZ]), derived from the ratio of PSA and transition zone volume (TZV).[11–14]

Since TRUS was introduced by Watanabe in 1967, its use to measure prostate volume has been important because of its improving image quality.[15] However, measurement of PV using TRUS is less accurate compared with MRI.[14, 16, 17] Further, MRI assesses PV with high reproducibility and accuracy compared with TRUS because of interobserver variability associated with the latter.[18]

Here we used logistic regression analysis and modeling to determine the efficacy of PSA levels, which were adjusted using MRI-based prostate zonal volume, and optimized models to differentiate PCa from BPH before initial prostate biopsy and for predicting HR-PCa among Chinese patients.

## Materials and methods

### Study population

This was a retrospective cohort study conducted in the Laboratory of Urology and the Department of Urology of Fujian Medical University Union Hospital (Fuzhou, China) from January 2012 to March 2018. Data were collected from 422 consecutive patients who underwent mp-MRI before initial TRUS-guided 13-core prostate biopsy. Patients met any of the criteria before initial prostate biopsy as follows: elevated PSA levels ($\geq 10$ ng·ml$^{-1}$), suspected cancer on digital rectal examination (DRE), hyperechoic or hypoechoic TRUS, or abnormal MRI findings. For PSA between 4 ng·ml$^{-1}$ to 10 ng·ml$^{-1}$, the biopsy criterion was free PSA<16% or PSAD >0.15 ng·ml$^{-2}$. We excluded patients with a history of prostate surgery or pathological examination revealing tumors other than adenocarcinoma. Ethical approval was acquired from the Institutional Review Board of Fujian Medical University Union Hospital. The approval form of consent was obtained by written with approval number of 2018KY078, and patients provided written informed consent before the study commenced. All data were fully anonymized before been accessed.

## Clinical date and variable definitions

Data on clinical characteristics including age, body mass index (BMI), PSA, percentage free PSA (free to total PSA), MR findings, PV, PSAD, TZV, PSADTZ, PZV, PSADPZ, alkaline phosphatase (ALP) and lactate dehydrogenase (LDH) were collected before biopsy. PV was calculated for each patient using the prolate ellipsoid formula (volume = 0.52 × length × width × height) using prostate T2WI MR images (axial and sagittal). MR imaging was performed using a 3.0T MR scanner (Siemens Medical Solutions, Erlangen, Germany). The interpretation of MRI findings was performed by a radiologist and a urologist to measure prostatic width and height on axial fat-saturated T2WI MR images and prostatic length on sagittal images (Fig 1). PZV = PV–TZV. and PSAD, PSADTZ and PSADPZ were calculated as ratios of PSA to total PV, TZV and PZV(PZV = PV—TZV), respectively.

Patients underwent standard TRUS-guided 13-core prostate biopsies. Four and two cores were acquired from the left PZ and left TZ, respectively, and four and two cores were acquired from the right PZ and right TZ, respectively. The last core was acquired depending on the imaging abnormalities. All biopsy specimens were reviewed by a pathologist to diagnose prostate cancer. According to the 2018 EAU clinical guidelines for prostate cancer, HR-PCa is defined as PSA $\geq$20 ng·ml$^{-1}$, with or without T stage $\geq$T2b, and with or without Gleason score $\geq$7. We selected these parameters to distinguish patients with or without HR-PCa.

## Statistical analysis

The values of continuous variables (Age, BMI, PSA, percentage free PSA, PV, PSAD, TZV, PSADTZ, PZV, PSADPZ, ALP and LDH) were not normally distributed. Therefore, Wilcoxon

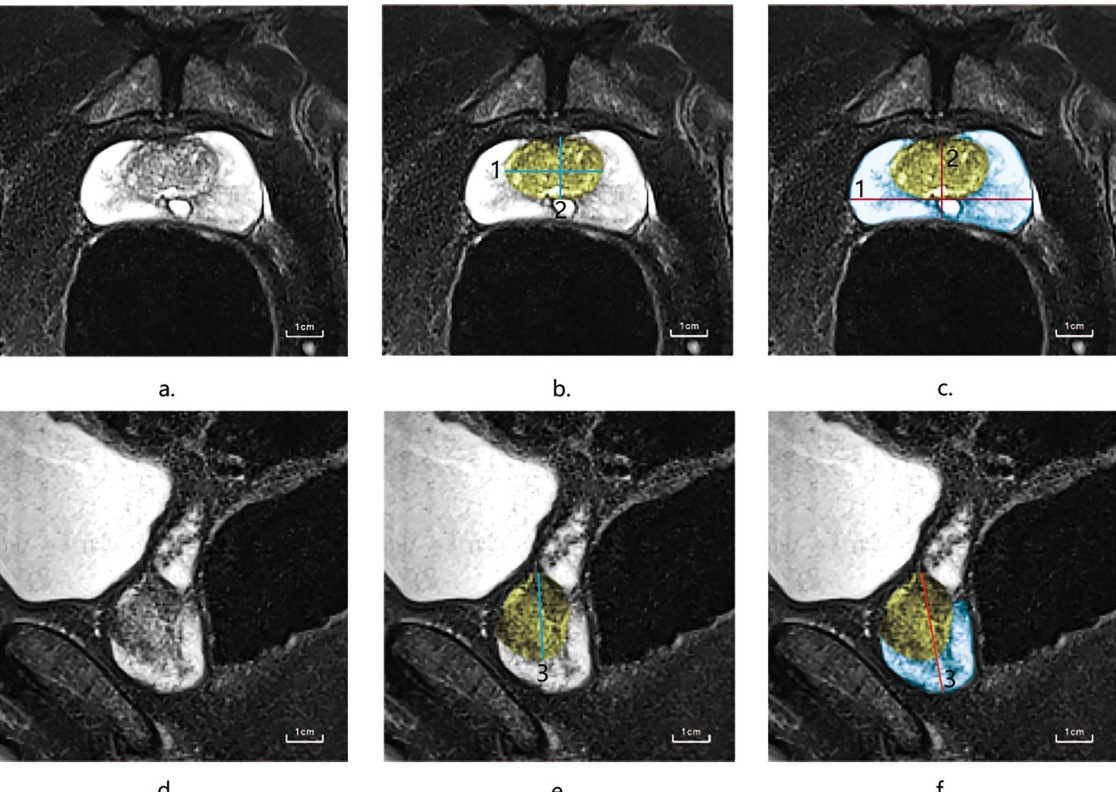

**Fig 1. Prostate location and MR imagine plane. (Yellow: Transition zone; Blue: Peripheral zone.).** (a, b, c.) Axial fat-saturated T2WI MR images of prostate. (d, e, f.) Sagittal fat-saturated T2WI MR images of prostate. (b, e.) Blue line (1, 2, 3.) depicting the width, height and length of transition zone. (c, f.) Red line (1, 2, 3.) depicting the width, height and length of prostate.

signed rank tests were used to evaluate these parameters, which are reported as the median with interquartile ranges. Categorical variables (MR findings) were calculated using the chi-squared test, shown as counts with percentages. Stepwise multivariate logistic regression analyses were performed to identify independent parameters associated with PCa and HR-PCa. We evaluated the diagnostic accuracy of base model 1 that integrated the clinical predictors PSA, MR, PV and PSAD. The logistic prediction models with TZV (base model 1 + TZV), with PSADTZ (base model 1 + PSADTZ), with PZV (base model 1 + PZV) and with PSADPZ (base model 1 + PSADPZ) were used to evaluate the biopsy results (with or without PCa). We evaluated the diagnostic accuracy of base model 2 that incorporated the clinical predictors MR, PV and PSAD as well as the logistic prediction models with TZV (base model 2 + TZV) and with PSADTZ (base model 2 + PSADTZ) to evaluate biopsy results (HR-PCa or no HR-PCa). The predictive accuracy of these variables and prediction models were calculated using the AUC. The cut-off value, sensitivity, specificity and positive and negative likelihood ratios were computed. Statistical significance was defined as P-value <0.05. Statistical analysis was performed using the Statistical Package for Social Sciences (SPSS version 20.0, Chicago, IL, USA).

## Results

Patients' baseline characteristics were summarized in Table 1. Of the 422 patients who underwent prostate biopsies, those of 185 (43.84%) were positive. Compared with patients with negative biopsies, the ages of patients with positive biopsies were significantly advanced and their values of PSA, PSAD, PSADTZ, PZV, PSADPZ, LDH and percentage of abnormal MR were higher as well. And lower PV and TZV values were found in patients with positive biopsies (each P <0.05). Among 185 patients, 71.35% (n = 132) were diagnosed with HR-PCa (Table 2). The ages of patients with HR-PCa were significantly advanced and their PV, PSAD,

**Table 1. Clinical characteristics of patients with no PCa and PCa at the initial biopsy.**

| Parameters | Overall | no PCa | PCa | OR (95% CI) | P value |
|---|---|---|---|---|---|
| Patients,n(%) | 422 | 237(56.16) | 185(43.84) | | |
| Age[a],years | 69(63–75) | 69(62.5–74) | 71(64–76) | 1.031(1.007–1.056) | 0.012 |
| BMI[a],kg·m$^{-2}$ | 23.40(21.50–25.50) | 23.30(21.73–24.98) | 23.50(20.80–25.80) | 1.001(0.943–1.062) | 0.560 |
| PSA[a], ng·ml$^{-1}$ | 13.56(8.62–31.18) | 11.50(7.55–18.54) | 26.10(11.34–97.90) | 1.039(1.029–1.050) | <0.001 |
| percent free PSA[a],% | 12.60(8.90–18.93) | 13.20(9.80–18.25) | 11.80(8.00–20.25) | 9.368(1.601–54.804) | 0.364 |
| MR,n(%) | | | | 5.616(3.409–9.253) | <0.001 |
| Normal | 132 (31.28) | 108 (45.57) | 24 (12.97) | | |
| Abnormal | 290 (68.72) | 129 (54.43) | 161 (87.03) | | |
| PV[a], ml | 52.41(38.99–71.97) | 59.37(41.97–82.65) | 45.05(34.02–59.83) | 0.978(0.970–0.986) | <0.001 |
| PSAD[a], ng·ml$^{-2}$ | 0.27(0.15–0.64) | 0.19(0.12–0.29) | 0.63(0.25–1.56) | 7.870(4.500–13.766) | <0.001 |
| TZV[a], ml | 19.94(10.84–36.88) | 32.14(17.43–48.50) | 12.35(7.56–19.30) | 0.926(0.910–0.943) | <0.001 |
| PSADTZ[a], ng·ml$^{-2}$ | 0.60(0.32–2.04) | 0.36(0.25–0.60) | 2.48(0.77–6.10) | 2.411(1.902–3.055) | <0.001 |
| PZV[a], ml | 28.96(23.02–38.25) | 27.79(21.59–35.81) | 32.90(23.94–42.24) | 1.027(1.012–1.043) | <0.001 |
| PSADPZ[a], ng·ml$^{-2}$ | 0.53(0.28–1.02) | 0.42(0.25–0.63) | 0.86(0.39–2.21) | 2.157(1.660–2.803) | <0.001 |
| ALP[a],U·L$^{-1}$ | 71(59–86) | 73(59–86) | 70(58–86.25) | 1.003(1.000–1.006) | 0.907 |
| LDH[a],U·L$^{-1}$ | 179(155–206) | 176(151–203) | 181(158–210) | 1.006(1.002–1.011) | 0.028 |

PCa prostate cancer, OR odds ratio, PV prostate volume, PSAD PSA density, TZV transitional zone volume, PSADTZ PSA density of the transitional zone, PZV peripheral zone volume, PSADPZ PSA density of the peripheral zone.

[a] Continuous variables are shown as the median value and interquartile range

**Table 2. Clinical characteristics of patients with no HR-PCa and HR-PCa at the initial biopsy.**

| Parameters | Overall | no HR-PCa | HR-PCa | OR (95% CI) | P value |
|---|---|---|---|---|---|
| Patients, n(%) | 185 | 53 (28.65) | 132 (71.35) | | |
| Age [a], years | 71(64–76) | 68(62.5–73.5) | 72(65–77) | 1.051(1.010–1.094) | 0.012 |
| BMI [a],kg·m$^{-2}$ | 23.5(20.8–25.8) | 24.4 (21.9–26.0) | 23.4 (20.7–25.7) | 0.932(0.848–1.023) | 0.139 |
| MR,n(%) | | | | 3.250(1.372–7.698) | 0.007 |
| Normal | 25 (13.51) | 13 (24.53) | 12 (9.09) | | |
| Abnormal | 160 (86.49) | 40 (75.47) | 120 (90.91) | | |
| PV [a], ml | 43.67 (32.51–59.07) | 40.62(25.22–56.46) | 46.28 (34.69–59.98) | 1.023(1.007–1.040) | 0.009 |
| PSAD [a], ng·ml$^{-2}$ | 0.76(0.29–1.74) | 0.30(0.19–0.46) | 1.09(0.45–1.90) | 2.214(1.406–3.487) | <0.001 |
| TZV [a], ml | 12.95 (7.60–20.48) | 16.42(9.38–25.42) | 11.62 (7.03–18.33) | 0.980(0.956–1.003) | 0.020 |
| PSADTZ [a], ng·ml$^{-2}$ | 3.05(0.98–7.06) | 1.05(0.48–2.41) | 3.83 (1.69–8.58) | 1.081(1.008–1.160) | <0.001 |
| PZV [a], ml | 32.90(23.94–42.24) | 28.25 (21.82–38.36) | 35.11 (25.33–43.81) | 1.033(1.006–1.061) | 0.015 |
| PSADPZ [a], ng·ml$^{-2}$ | 1.06(0.46–2.21) | 0.45(0.27–0.69) | 1.59(0.65–2.45) | 4.691(2.529–8.533) | <0.001 |
| ALP [a], U·L$^{-1}$ | 70(58–86.25) | 64(55–75) | 76(61–94) | 1.026(1.008–1.044) | 0.001 |
| LDH [a], U·L$^{-1}$ | 181(158–210) | 174(156–202) | 185.5 (158.25–214.5) | 1.009(1.000–1.018) | 0.095 |

HR-PCa high-risk prostate cancer, OR odds ratio, PV prostate volume, PSAD PSA density, TZV transitional zone volume, PSADTZ PSA density of the transitional zone, PZV peripheral zone volume, PSADPZ PSA density of the peripheral zone.

[a] Continuous variables are shown as the median value and interquartile range

PSADTZ, PZV and PSADPZ values were higher as well. Their LDH and TZV values were lower compared with those of patients without HR-PCa (each P <0.05).

Multivariate logistic regression analysis revealed that PSA, MR, PV, PSAD, TZV, PSADTZ, PZV and PSADPZ served as independent predictors of PCa (Table 3). Further, multivariate logistic regression analysis incorporating backward elimination selection was used to select independent predictors of HR-PCa in model building. Stepwise multivariate analysis that excluded Age, PZV, PSADPZ and LDH revealed that MR, PV, PSAD, TZV and PSADTZ were independent predictors of HR-PCa (Table 4).

Tables 5 and 6 show the ROC curve analysis of the different clinical parameters of PCa and HR-PCa. When we chose the best cut-off values of PSA, PSADTZ, PSADPZ (29.165, 0.705 and 0.975, respectively) for predicting PCa, the sensitivities were 48.9%, 78.3% and 48.4%, respectively and the specificities were 91.5%, 81.4% and 89.8%, respectively. The best cut-off values for predicting HR-PCa using PV, TZV, PSADTZ were 28.18, 19.23 and 1.658, respectively; sensitivities were 92.4%, 78.0% and 75.6%, respectively; and specificities were 35.8%, 56.6% and 28.3%, respectively.

Base model 1 for PCa integrated PSA, PV, PSAD and MR. The AUC of base model 1 + TZV or base model 1 + PZV for PCa was higher compared with those of PSA alone, base model 1, base model 1 + PSADTZ, or base model 1 + PSADPZ. The AUC of base model 1 + TZV was 0.906 for PCa, which was similar to that of base model 1 + PZV. Base model 2 for HR-PCa integrated PV, PSAD and MR. The AUC of base model 2 + TZV was 0.893 for HR-PCa, which was higher compared with that of base model 2 or base model 2 + PSADTZ (Fig 2).

## Discussion

Definitive diagnosis of prostate cancer depends on histopathological verification of adenocarcinoma in a prostate biopsy that may lead to complications such as infection, bleeding and anxiety.[3] Therefore, it is inappropriate to use a prostate biopsy to perform routine active

**Table 3. Multivariate logistic regression analysis of predictors associated with PCa at the initial biopsy.**

| Parameters | Multivariate analysis adjusted OR for PCa | Multivariate analysis adjusted 95% CI for PCa | P value |
|---|---|---|---|
| Age | 1.025 | 0.989–1.061 | 0.171 |
| PSA | 1.044 | 1.018–1.072 | 0.001 |
| MR | 2.856 | 1.455–5.608 | 0.002 |
| PV | 1.077 | 1.044–1.112 | <0.001 |
| PSAD | 0.018 | 0.001–0.330 | 0.007 |
| TZV | 0.841 | 0.795–0.890 | <0.001 |
| PSADTZ | 2.494 | 1.158–5.370 | 0.020 |
| PZV | 1.189 | 1.124–1.258 | <0.001 |
| PSADPZ | 2.016 | 1.147–3.544 | 0.015 |
| LDH | 1.001 | 0.994–1.009 | 0.688 |

PCa prostate cancer, OR odds ratio, PV prostate volume, PSAD PSA density, TZV transitional zone volume, PSADTZ PSA density of the transitional zone, PZV peripheral zone volume, PSADPZ PSA density of the peripheral zone.

surveillance on asymptomatic people with elevated serum levels of PSA.[2] Reducing the complications of biopsies requires careful selection of patients who are likely to benefit. Although a series of parameters and imaging methods are available to improve patient selection, there is no consensus on the optimal criteria. To address the situation, here we evaluated the reliability of PV and associated zone-adjusted PSA levels for detecting prostate cancer.

Measurements of PSA level in serum are considered to help detect prostate cancer.[2] However, the elevation of PSA levels may be caused by BPH rather than prostate cancer.[19] Consequently, numerous studies report the predictive value of PV-adjusted PSA for PCa. For example, a study of the utility of PSAD and PSA of 659 patients demonstrated that the AUC of PSAD (0.73) is higher than that of PSA (0.61) for diagnosing PCa.[7] A study of 172 patients found that the AUC of PSA significantly increases from 0.683 to 0.806 using PSAD.[12] However, a study of 109 patients with clinically localized prostate cancer found that PSAD fails to outperform PSA for preoperative prediction of prostate cancer.[20]

We were unable to determine the reasons for the dissimilarities among these studies. However, measurement of prostate volume may be less accurate using TRUS compared with mp-MRI or specimen after radical prostatectomy. To increase the accuracy of our findings, we used mp-MRI-based parameters and found a higher AUC of PSAD (0.794) than that of PSA (0.723). Multivariate regression analysis for predicting PCa indicated that PSAD was superior to PSA for making decisions on selecting patients to undergo biopsy.

Here we found patients with PCa had lower PV compared with those without PCa (45.05 vs 59.37 ml, P <0.001). We assumed that BPH might contribute to the increase of PV to a greater

**Table 4. Multivariate logistic regression analysis of predictors associated with HR-PCa at the initial biopsy (backward elimination selection procedure).**

| Parameters | Multivariate analysis adjusted OR for HR-PCa | Multivariate analysis adjusted 95% CI for HR-PCa | P value |
|---|---|---|---|
| MR | 3.576 | 1.095–11.683 | 0.035 |
| PV | 1.154 | 1.094–1.217 | <0.001 |
| PSAD | 106.450 | 12.424–912.037 | <0.001 |
| TZV | 0.810 | 0.742–0.883 | <0.001 |
| PSADTZ | 0.708 | 0.549–0.915 | 0.008 |

HR-PCa high-risk prostate cancer, OR odds ratio, PV prostate volume, PSAD PSA density, TZV transitional zone volume, PSADTZ PSA density of the transitional zone.

**Table 5. The AUC and cut-off values for predicting biopsy outcome and their sensitivity, specificity, positive and negative likelihood ratios for PCa and no PCa.**

| Parameters | AUC (95% CI) | Cut-off value | Sensitivity (%) | Specificity (%) | Positive likelihood ratio | Negative likelihood ratio |
|---|---|---|---|---|---|---|
| PSA | 0.723(0.672–0.774) | 29.165 | 48.9 | 91.5 | 0.8179 | 0.6964 |
| MR | 0.661(0.610–0.713) | — | — | — | — | — |
| PV | 0.339(0.287–0.391) | 52.73 | 65.4 | 60.8 | 0.5657 | 0.6924 |
| PSAD | 0.794(0.749–0.839) | 0.365 | 65.8 | 83.9 | 0.7614 | 0.7586 |
| TZV | 0.197(0.155–0.238) | 19.415 | 76.2 | 27.8 | 0.4517 | 0.5944 |
| PSADTZ | 0.862(0.826–0.899) | 0.705 | 78.3 | 81.4 | 0.7667 | 0.8277 |
| PZV | 0.607(0.552–0.661) | 36.80 | 40.2 | 80.1 | 0.6119 | 0.6318 |
| PSADPZ | 0.704(0.652–0.756) | 0.975 | 48.4 | 89.8 | 0.7874 | 0.6903 |
| Base model 1 | 0.842(0.803–0.881) | 0.513 | 67.4 | 88.6 | 0.8219 | 0.7769 |
| Base model 1+ TZV | 0.906(0.879–0.934) | 0.422 | 85.3 | 80.5 | 0.7735 | 0.8752 |
| Base model 1+ PSADTZ | 0.880(0.846–0.914) | 0.354 | 80.4 | 83.5 | 0.7918 | 0.8451 |
| Base model 1+ PZV | 0.906(0.879–0.934) | 0.422 | 85.3 | 80.5 | 0.7735 | 0.8752 |
| Base model 1+ PSADPZ | 0.851(0.813–0.888) | 0.459 | 69.6 | 87.3 | 0.8105 | 0.7863 |

PCa prostate cancer, AUC area under the receiver-operating characteristic curve, PV prostate volume, PSAD PSA density, TZV transitional zone volume, PSADTZ PSA density of the transitional zone, PZV peripheral zone volume, PSADPZ PSA density of the peripheral zone, Base model 1 PSA + PV + PSAD + MR.

extent than PCa. PCa usually arises from the PZ, and most BPH originates in the TZ.[10, 12] Our result are consistent with this opinion. Compared with patients without PCa, lower TZV (12.35 vs 32.14 ml, P <0.001) and higher PZV (32.9 vs 27.79 ml, P <0.001) was found in patients with PCa, which indicates the diagnostic potential of TZV and PZV for diagnosing PCa.

PSA mainly leaks from the TZ.[10, 12] Therefore, we hypothesized that prostate-zone adjusted PSA serves as a more effective parameter than PSAD and PSA for the diagnosis of PCa, which is consistent with published studies. For example, a study of 1712 patients who underwent TRUS-guided prostate biopsies found that the AUC of PSADTZ is 0.766 and those of PSAD and PSA are 0.749 and 0.585, respectively, for diagnosing PCa.[21] Another study of 189 patients stratified according to PSA levels ranging from 4.0–10.0 ng·ml$^{-1}$ or 10.1–20.0 ng·ml$^{-1}$ found that the AUC of PSADTZ were higher (0.702 and 0.730, respectively), compared with those of PSA (0.569 and 0.463, respectively) as well as the AUC associated with specificity and sensitivity.[22] Here, we found that the AUC of PSADTZ (0.862) was significantly higher

**Table 6. The AUC and cut-off values for predicting biopsy outcome and their sensitivity, specificity, positive and negative likelihood ratios for HR-PCa and no HR-PCa.**

| Parameters | AUC (95% CI) | Cut-off value | Sensitivity(%) | Specificity(%) | Positive likelihood ratio | Negative likelihood ratio |
|---|---|---|---|---|---|---|
| MR | 0.577(0.482–0.672) | — | — | — | — | — |
| PV | 0.625(0.531–0.720) | 28.18 | 92.4 | 35.8 | 0.7819 | 0.6542 |
| PSAD | 0.732(0.642–0.823) | 0.56 | 72.5 | 18.9 | 0.6900 | 0.2163 |
| TZV | 0.393(0.301–0.485) | 19.23 | 78.0 | 56.6 | 0.8174 | 0.5081 |
| PSADTZ | 0.729(0.639–0.818) | 1.658 | 75.6 | 28.3 | 0.7242 | 0.3177 |
| Base model 2 | 0.809(0.748–0.870) | 0.8347 | 54.2 | 69.2 | 0.8142 | 0.3777 |
| Base model 2 + TZV | 0.893(0.849–0.937) | 0.8130 | 70.2 | 98.1 | 0.9892 | 0.5693 |
| Base model 2 + PSADTZ | 0.814(0.755–0.874) | 0.8197 | 58 | 1.9 | 0.5955 | 0.0178 |

HR-PCa high-risk prostate cancer, AUC area under the receiver-operating characteristic curve, PV prostate volume, PSAD PSA density, TZV transitional zone volume, PSADTZ PSA density of the transitional zone, Base model 2 PV + PSAD + MR.

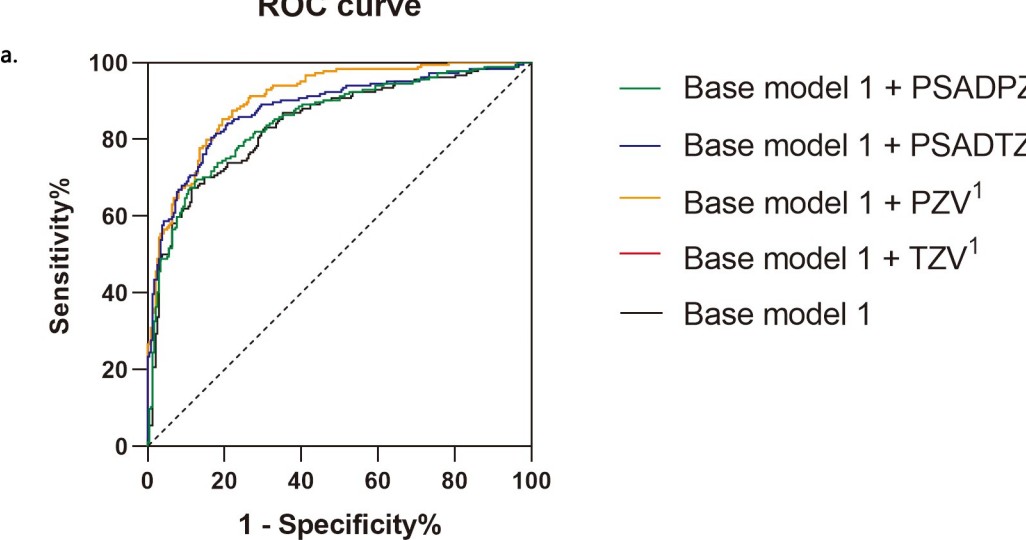

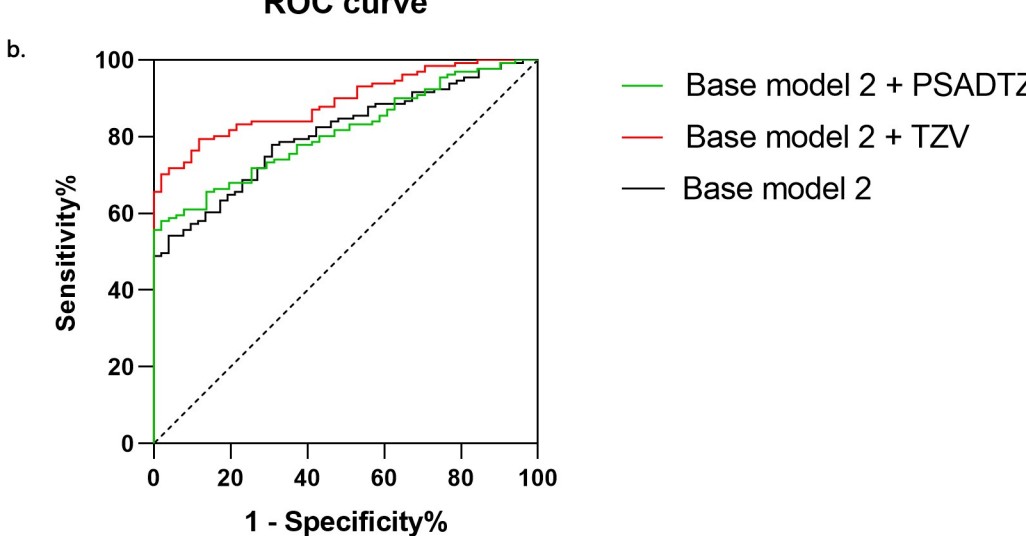

**Fig 2. Receiver-operating characteristic curves depicting the accuracy of predictors of PCa (a.) and HR-PCa (b.) at the initial biopsy.** Base model 1, PSA + PV + PSAD + MR; Base model 2, PV + PSAD + MR. [1]Base model 1 + TZV and Base model 1 + PZV had the same AUC.

compared with that of PSA (0.723) or that of the AUC of any other single parameter, followed by the AUC of PSAD (0.794). Further, we show here that our data provide a compelling argument that supports the conclusion that the utility of PSADTZ for performing surveillance for patients at risk of PCa is more effective compared with standard variables such as PSA.

Dissimilarities between single parameters are occasionally reported, and models that integrate multiple parameters were developed to predict prostate cancer more accurately. For

example, a model developed based on 862 men who underwent TRUS found that the highest AUC (0.905) was associated with their best model that integrates age, PSA, percentage free PSA, PV, DRE and TURS, which was higher compared with that of PSA alone (0.672). [23] This study further evaluated the predictive accuracy of the best models (0.90 in the internal validation). Another study integrated PSAD and the percentage free PSA as their best model (AUC = 0.824) for the probability of detecting prostate cancer in all patients; and the AUC for PSA, percentages of free PSA and PSAD were 0.662, 0.676 and 0.786 respectively.[24] To validate the utility of their model, the best model was applied to an independent cohort of 88 patients. The results showed its AUC is 0.883, which is greater than those of PSA (0.704) and PSAD (0.854) for predicting PCa in the test cohort.

In the present study, we integrated PSA, PV, PSAD and MR as base model 1. When integrated with TZV or PZV, these variables equally served as the best model (AUC = 0.906 for predicting PCa). Interestingly, the AUC associated with PSADTZ was the highest among all single parameters, but when integrated into the base model 1, its AUC is 0.880, which is lower compared with that of the base model 1 + TZV or PZV. We assumed that integrated models eliminated confounding factors of single parameter. We concluded therefore that our best models are superior to those of the studies cited.

As described by the International Society of Urological Pathology 2014 grade, patients diagnosed with PCa are stratified using the combination of serum PSA level, Gleason score and clinical staging (cTNM).[2] Risk stratification can help clinicians select treatment strategies such as curative or deferred treatment and predict the outcomes of patients with different levels of risk. However, risk stratification relies on prostate biopsy which may cause the complications mentioned above. Therefore, we conclude that a novel method based on noninvasive parameters may be superior to prostate biopsy for risk stratification.

In the present study, compared with no HR-PCa, patients diagnosed with HR-PCa had higher PV (46.28 vs 40.62 ml, P <0.001) and PZV (35.11 vs 28.25 ml, P <0.001), possibly indicating the higher rate of tumor growth and pathological progression, which more likely may be stratified as HR-PCa, which corresponds with the origin of most PCa in the PZ. For example, one study found that 44% of 380 patients had extracapsular extensions, indicating that PSAD is an independent predictor that distinguishes HR-PCa from PCa and predicts PSA-free survival.[25] Further, the 5-year PSA-free survival rates are 82.9% for patients with PSAD <0.468 ng·ml$^{-2}$ and 50.7% for those with PSAD >0.468 ng·ml$^{-2}$ (P <0.001). Here we show that PSAD had the highest AUC (0.732) with a cut-off value of 0.56 ng·ml$^{-2}$, followed by the AUC of PSADTZ (0.729), which distinguished HR-PCa from PCa. We assumed that the discrepancy in the PSAD cut-off values may be explained by different imaging techniques used to measure PV. We believe it is therefore reasonable to conclude that PSAD and PSADTZ may serve as optimal noninvasive parameters when applied to evaluate risk stratification among patients with PCa.

Other models are available to predict HR-PCa. For example, a study of 362 patients with PCa developed a predictive model (AUC = 0.894) by integrating age, PSA, PV, DRE and TRUS.[23] Another study of 216 patients PCa included 97 patients with HR-PCa. Age, PSA, percentage free PSA, PV, DRE and TRUS were integrated into one model with an AUC of 0.830 for predicting HR-PCa.[26] The discrepant combinations of models in many reports differ widely.

In the present study, we used PZV and zone-adjusted PSA based on mp-MRI rather than TRUS to build models which is different from those developed by other investigations. We found that TZV and PSADTZ were crucial contributors to our prediction model (AUC = 0.893 and 0.814, respectively), which were simultaneously integrated into our base model 2 (PV+PSAD+MR). We were unaware of a consensus opinion for determining an

absolute superior combination of parameters in predictive models. Because predictive models were built using different populations of patients and different analytical methods, we assumed that our model integrating TZV, PV, PSAD and MR is superior to those developed for other combinations of variables.

Our study has certain limitations. Except for the small sample size, our models were not calibrated using internal and external validation to ensure their utility before being applied to patients. Further studies of clinical practice are therefore required that employ long-term follow-up to evaluate the applicability of our model.

## Conclusion

MRI-based PSADTZ and PSAD have potential predictive value for diagnosing PCa and differentiating patients with or without HR-PCa. The application of base models integrated with PZV, TZV and PSADTZ may further improve the predictive accuracy of the diagnosis of PCa and HR-PCa. MRI is used widely in clinical practice. MRI-based model can help clinicians avoid performing unnecessary prostate biopsies and evaluating risk stratification of prostate cancer.

## Supporting information

**S1 File. Raw data.**
(ZIP)

## Acknowledgments

We thank Liwen Bianji, Edanz Editing China (www.liwenbianji.cn/ac), for editing the English text of a draft of this manuscript.

## Author Contributions

**Conceptualization:** Shaoqin Jiang, Mengqiang Li.

**Data curation:** Zhenlin Chen, Wenzhen Shi, Yue Xu, Yinan Guo.

**Formal analysis:** Zhenlin Chen, Zhangcheng Huang, Wenzhen Shi, Bingqiao Liu, Huijie Yang.

**Funding acquisition:** Mengqiang Li.

**Supervision:** Mengqiang Li.

**Validation:** Song Zheng.

**Writing – original draft:** Shaoqin Jiang, Zhangcheng Huang.

**Writing – review & editing:** Song Zheng.

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
