## [Decision Letter · Decision Letter 0]

10 Sep 2019

PONE-D-19-16015

The roles of MRI-based prostate volume and associated zone-adjusted prostate-specific antigen concentrations in predicting prostate cancer and high-risk prostate cancer

PLOS ONE

Dear Dr. Li,

Thank you for submitting your manuscript to PLOS ONE. After careful consideration, we feel that it has merit but does not fully meet PLOS ONE’s publication criteria as it currently stands. Therefore, we invite you to submit a revised version of the manuscript that addresses the points raised during the review process.

We would appreciate receiving your revised manuscript by Oct 25 2019 11:59PM. To enhance the reproducibility of your results, we recommend that if applicable you deposit your laboratory protocols in protocols.io, where a protocol can be assigned its own identifier (DOI) such that it can be cited independently in the future. For instructions see: http://journals.plos.org/plosone/s/submission-guidelines#loc-laboratory-protocols

We look forward to receiving your revised manuscript.

Kind regards,

Isaac Yi Kim, MD, PhD

Academic Editor

PLOS ONE

Journal Requirements:

2. In the ethics statement in the manuscript and in the online submission form, please provide additional information about the patient records used in your retrospective study. Specifically, please ensure that you have discussed whether all data were fully anonymized before you accessed them.

Reviewers' comments:

Reviewer's Responses to Questions

**Comments to the Author**

1. Is the manuscript technically sound, and do the data support the conclusions?

Reviewer #1: Yes

Reviewer #2: Yes

2. Has the statistical analysis been performed appropriately and rigorously? 

Reviewer #1: Yes

Reviewer #2: Yes

3. Have the authors made all data underlying the findings in their manuscript fully available?

Reviewer #1: Yes

Reviewer #2: Yes

4. Is the manuscript presented in an intelligible fashion and written in standard English?

Reviewer #1: Yes

Reviewer #2: Yes

5. Review Comments to the Author

Reviewer #1: Interesting evaluation of MRI to predict prostate cancer.

Major Points:

- Please elaborate on the 13 core biopsy technique you used. I am more familiar with a 12 core sampling of the peripheral zone. Some urologists also perform anterior biopsies for a total of 14 cores. Targeted biopsies based on multiparametric MRI findings are then done.

- Were your targeted biopsies "cognitive" or did you use MRI-U/S fusion technology such as Artemis or UroNav?

- What percentage of standard biopsies were positive for cancer vs. those targeting a MRI lesion?

Minor Points:

- Please format the references according to journal specifications. Numerous citations were missing author names.

Reviewer #2: Dear Authors:

Please clarify - Ethics Statement on Page 3 is “N/A” and the “Materials and Methods” states “Ethics approval was acquired from the IRB…” Please adjust the Ethics statement accordingly.

Please clarify: “Material and Methods” states this was “retrospective cohort study” and the “Clinical Date and Variable Definition” states ALP and LDH were collected before biopsy. Was that part of the study or was it standard of care. This is important due to the concern above regarding ethics statement.

If you feel appropriate, I would like your thoughts on the existing PIRADs classification used in clinical practice and how it relates to your findings.

Editorial comments:

Consider defining abbreviations in the abstract (PV, PSAD, TZV etc.)

6. PLOS authors have the option to publish the peer review history of their article (what does this mean?). If published, this will include your full peer review and any attached files.

Reviewer #1: No

Reviewer #2: Yes: Biren Saraiya

---

## [Author Response · Author response to Decision Letter 0]

19 Oct 2019

Thank you for your letter and for the reviewers’ comments concerning our manuscript entitled “The roles of MRI-based prostate volume and associated zone-adjusted prostate specific antigen concentrations in predicting prostate cancer and high-risk prostate cancer” (Manuscript number: PONE-D-19-16015). Those comments are all valuable and very helpful for revising and improving our paper, as well as the important guiding significance to our researches. We have studied comments carefully and have made correction which we hope meet with approval. Revised portion are highlighted in a marked-up copy of my manuscript. The main corrections in the paper and the responds to the reviewer’s comments are as following:

Responds to the reviewer’s comments:

Reviewer #1: 

1. Please elaborate on the 13 core biopsy technique you used. I am more familiar with a 12 core sampling of the peripheral zone. Some urologists also perform anterior biopsies for a total of 14 cores. Targeted biopsies based on multiparametric MRI findings are then done.

Response: Thank you for suggesting us to make a more clear statement of the 13-core biopsy technique that we used in our clinical work. As we mentioned in our manuscript, four and two cores were acquired from the left PZ and left TZ, respectively, and four and two cores were acquired from the right PZ and right TZ, respectively. The last core was examined for abnormalities found in imageological examination. The number of biopsy cores still remains inconclusive. In our routine clinical work, we perform standard TRUS-guided 13-core prostate biopsy for patient.

2. Were your targeted biopsies "cognitive" or did you use MRI-U/S fusion technology such as Artemis or UroNav?

Response: We are sorry for our negligence of the method of our targeted biopsies. We did use cognitive fusion technology in the last core of biopsy.

3. What percentage of standard biopsies were positive for cancer vs. those targeting a MRI lesion?

Response: Thank you for putting forward such a valuable question. The last core, so called cognitive biopsy, was a part of systematic biopsies. We didn’t compare the positve rate of each core. 

4. Please format the references according to journal specifications. Numerous citations were missing author names.

Response: As Reviewer suggested, our software had something wrong in citing references. But we have fixed it and made correction according to Endnote style file, named “PLoS (Public Library of Science – all journals)”, downloaded from PLOS ONE.

Special thanks to you for your good comments. 

Reviewer #2: 

1. Please clarify - Ethics Statement on Page 3 is “N/A” and the “Materials and Methods” states “Ethics approval was acquired from the IRB…” Please adjust the Ethics statement accordingly.

Response: It is really true as Reviewer suggested that we missed the ethics approval statement. Ethical approval was acquired from the Institutional Review Board of Fujian Medical University Union Hospital. The approval form of consent was obtained by written with approval number of 2018KY078. Now we have submit the ethics approval statement in submission system.

2. Please clarify: “Material and Methods” states this was “retrospective cohort study” and the “Clinical Date and Variable Definition” states ALP and LDH were collected before biopsy. Was that part of the study or was it standard of care. This is important due to the concern above regarding ethics statement.

Response: Thank you for indicating the potential ethics problem. We routinely perform biochemical analysis on patients, which including ALP and LDH. So it is standard of care for patients.

3. If you feel appropriate, I would like your thoughts on the existing PIRADs classification used in clinical practice and how it relates to your findings.

Response: Thank you for giving us a hint that will provide a good perspective for improving clinical practice. PIRADs classification is a well-recognized technique that can improve diagnostic performance in prostate cancer. Now we are still working on the PI-RADs v2. And I believe we will get acquainted with it in our next research.

Special thanks to you for your good comments. 

Responds to editor comments:

1.We have ensured that our manuscript meets PLOS ONE's style requirements through PLOS ONE style templates. If our manuscript still exists any problem, please tell me. 

2.The information of patients which we used in our retrospective study has been mentioned in the Clinical data and variable definitions part in our manuscript. We have ensured that all data were fully anonymized before we accessed them.

3.Considering the words limitation of abstract, so we didn’t define abbreviations in the abstract. But as editor suggested, now we have defined abbreviations in the abstract, including PCa, PV, PSAD, TZV, PSADTZ, PZV and PSADPZ.

We appreciate for Editors/Reviewers’ warm work earnestly, and hope that the correction will meet with approval.

Once again, thank you very much for your comments and suggestions.

---

## [Editor Report · Decision Letter 1]

23 Oct 2019

PONE-D-19-16015R1

The roles of MRI-based prostate volume and associated zone-adjusted prostate-specific antigen concentrations in predicting prostate cancer and high-risk prostate cancer

PLOS ONE

Dear Dr. Li,

Thank you for submitting your manuscript to PLOS ONE. After careful consideration, we feel that it has merit but does not fully meet PLOS ONE’s publication criteria as it currently stands. Therefore, we invite you to submit a revised version of the manuscript that addresses the points raised during the review process.

1. "Imageoloical" is not a standard word used to describe MRI or other imagings. Please revise the relevant sentence appropriately,.

We would appreciate receiving your revised manuscript by Dec 07 2019 11:59PM. To enhance the reproducibility of your results, we recommend that if applicable you deposit your laboratory protocols in protocols.io, where a protocol can be assigned its own identifier (DOI) such that it can be cited independently in the future. For instructions see: http://journals.plos.org/plosone/s/submission-guidelines#loc-laboratory-protocols

We look forward to receiving your revised manuscript.

Kind regards,

Isaac Yi Kim, MD, PhD

Academic Editor

PLOS ONE

---

## [Author Response · Author response to Decision Letter 1]

25 Oct 2019

Dear Editors:

Thank you for your letter concerning our manuscript entitled “The roles of MRI-based prostate volume and associated zone-adjusted prostate specific antigen concentrations in predicting prostate cancer and high-risk prostate cancer” (Manuscript number: PONE-D-19-16015). This comment is all valuable and very helpful for revising and improving our paper, as well as the important guiding significance to our researches. We have studied it carefully and have made correction which we hope meet with approval. Revised portion are highlighted in a marked-up copy of my manuscript. The main correction in the paper and the responds is as following:

1. 1. "Imageoloical" is not a standard word used to describe MRI or other imagings. Please revise the relevant sentence appropriately.

Response: Thank you for indicating our improper use of words. We have corrected this sentence to make it concise and to the point.

We appreciate for Editors’ warm work earnestly, and hope that the correction will meet with approval.

Once again, thank you very much for your comments and suggestions.

Sincerely,

Mengqiang Li

---

## [Editor Report · Decision Letter 2]

30 Oct 2019

The roles of MRI-based prostate volume and associated zone-adjusted prostate-specific antigen concentrations in predicting prostate cancer and high-risk prostate cancer

PONE-D-19-16015R2

Dear Dr. Li,

We are pleased to inform you that your manuscript has been judged scientifically suitable for publication and will be formally accepted for publication once it complies with all outstanding technical requirements.

With kind regards,

Isaac Yi Kim, MD, PhD

Academic Editor

PLOS ONE
---

## [Editor Report · Acceptance letter]

6 Nov 2019

PONE-D-19-16015R2 

The roles of MRI-based prostate volume and associated zone-adjusted prostate-specific antigen concentrations in predicting prostate cancer and high-risk prostate cancer 

Dear Dr. Li:

I am pleased to inform you that your manuscript has been deemed suitable for publication in PLOS ONE. Congratulations! Your manuscript is now with our production department. 

With kind regards,

on behalf of

Dr. Isaac Yi Kim 

Academic Editor

PLOS ONE